# Training Teachers for the Career Guidance of High School Students

**Otilia Sanda Bersan [1], Anca Lustrea [2], Simona Sava [2,*] and Oana Bobic [3]**

1   Department of Educational Sciences, West University of Timisoara, 300223 Timisoara, Romania; otilia.bersan@e-uvt.ro
2   Department of Educational Sciences, University Clinic of Therapies and Psycho-Pedagogical Counseling, West University of Timisoara, 300223 Timisoara, Romania; anca.lustrea@e-uvt.ro
3   Doctoral School of Pshychology-Educational Sciences, West University of Timisoara, 300223 Timisoara, Romania; oana.batrina10@e-uvt.ro
*   Correspondence: lidia.sava@e-uvt.ro

**Abstract:** The article addresses the decision-making process of career choice among high-school students and emphasizes the importance of supporting their informed decisions by trained teachers acting as career-guidance counselors. While, ideally, school counselors handle career counseling, their limited availability necessitates the involvement of other resources, such as trained teachers. The present study introduces a career-guidance training program for teachers, implemented with 20 Romanian and 20 Serbian teachers. The research conducted simultaneously with the training aimed to assess the effectiveness of this cross-national program in enhancing teachers' competence in career guidance. Utilizing a longitudinal mixed methodology, the study assessed the teachers' perceptions of the training's effectiveness and sustainability over a period of 24 months. Two questionnaires featuring multiple-choice and open-choice questions were employed. The results consistently indicated that teachers rated the training as excellent or very good across various dimensions, including content, trainers, didactic materials, and applications. Challenges were noted in designing and implementing group career activities compared to individual ones, with no significant differences observed between Serbian and Romanian teachers. After 24 months, a deductive content analysis of open-ended questions assessed the sustainability of acquired competencies. Our findings indicated active teacher participation in career-guidance activities, primarily with final-year students serving as class teachers or subject instructors. In the context of a scarcity of career-counseling specialists, training teachers as career-guidance advisors emerges as a viable solution. The study highlights the potential of such training programs to address the critical need for comprehensive career guidance in schools.

**Keywords:** career guidance; teacher training; high-school students; teachers as career-guidance counselors

## 1. Introduction

Career guidance is a process that assists individuals in making informed decisions about their career paths by providing information, support, and resources to help them explore and navigate various career options [1]. It involves activities and programs that integrate knowledge, experience, and insights that promote self-knowledge, an understanding of societal dynamics, and to engage in career decision-making processes.

Career guidance is a complex process involving various disciplinary and theoretical approaches [2,3]. The first major career transition, from high school to work, significantly impacts future development. Research on career counseling and the school-to-work career transition focused on individual, agentic perspectives and institutional arrangements [3].

Given the restricted number of specialized support services, and the fact that there are too few counselors in schools, career guidance in Romania is limited because un-

trained teachers carry out the guidance activities [4]. Teachers often avoid providing career guidance due to feeling unprepared [5].

The INTERREG-IPA CBC Romania–Serbia project, RORS 406, titled "e-Support services for career and vocational counseling of youth entering to the labor market", conducted from 2021 to 2022, sought to address the shortage of counseling services by training high-school teachers in career guidance. This transnational and cross-border initiative emerged from the shared necessity for enhanced career-support services and the exchange of cross-border best practices in the field. The Serbian part of Banat has a large number of Romanian minorities, and many Romanian students from there pursue academic careers in Romania, benefiting from special incentives.

This paper presents this training solution aimed at enabling high-school teachers to provide career guidance, illustrating their self-perceived capabilities and challenges in providing career guidance, after completing the training.

The training and research draw on the socio-cognitive (career) theory (SCCT) [6], building on the idea that the social contexts the teachers are exposed to during the training inspire and enable them to conduct in the classroom the training situations they have experienced. It implies that people's observations of other people, their own experiences, and the social context in which they operate all have an impact on their capabilities [6]. The SCCT also offered a framework for the curricular design of teacher training, in addition to the classic career-construction theory. According to the SCCT, there is a dynamic interaction between behavioral, environmental, and personal factors that affect career development [6]. SCCT emphasizes the importance of self-efficacy beliefs, outcome expectations, and goals in shaping individuals' career choices and actions. Based on SCCT and the career-construction theory, the theoretical component of the teacher training covered topics such as personal development, communication, learning management, and career planning. Teachers were instructed to help students discover their abilities, recognize strengths, and form evidence-based self-efficacy beliefs to inform career choices. Additionally, topics related to lifestyle and the labor market equipped teachers to assist students in aligning economic requirements with personal preferences. Practical activities, overseen by experts, enabled teachers to guide students in analyzing self-efficacy beliefs, professional environments, and personal factors to make informed career decisions.

Through exercises, observations, reflections, and systematic exposure to career-guidance training, teachers became motivated to develop and solidify career-guidance skills. They implemented career-guidance activities with their students, receiving support from counselors and an expert team during the RORS 406-dedicated project.

*1.1. Career Guidance in Romania and Serbia*

National reports from Romania and Serbia on career counseling highlighted some common and culturally specific barriers in both countries. In Romania, the major barrier is the deficit of specialized career counselors and the need for interventions at both systemic and school levels to enhance career-advisory and counseling (CAC) activities [4]. The reports emphasized the importance of motivating students and parents and making them aware of the role of the baccalaureate exam and CAC activities in career development [4,7]. The high unemployment rates among youth in Romania and Serbia further underscored the need for proactive CAC services, such as pre-transition preparation from school to work or further education. Training additional teachers, especially class teachers, in career guidance was suggested as a potential solution [4,7]. Romanian schools are planning to have at least one specialist counselor per 800–1000 students, but in reality, the number of students to deal with can increase to 1300 [4,8]. CAC services are also needed to address the 5.7% youth unemployment rate in Romania and 10.9% in Serbia in 2022 [9], with a large percentage being long-term unemployed and not in education or training—NEETs [10]. Low enrollment in academic programs, high academic-dropout rates, and unhappiness at work are the main causes of a lack of CAC, as studies in Romania underlined [8].

In Serbia, career guidance in schools is well stipulated in the policy documents, such as the education law on secondary education (2013, 2017) [11] and the *National Education Strategy* (2021–2030) [12]. Career guidance is seen as an important provision for individuals with well-established standards [13]. However, the teachers in schools are less confident in providing career guidance, even though they are supposed to implement predefined school programs as part of a team, perceiving more training needs in this regard, as the Serbian teachers attending the training indicated.

### 1.2. Career Guidance for High-School Students

Navigating a career in a rapidly changing and unpredictable environment is a complex and ongoing process, characterized by challenges and sometimes interrupted by uncertain job opportunities [2]. Career guidance services need to align with this ever-shifting occupational terrain, offering diverse support services and guidance activities to optimize human potential, foster self-realization, and enhance alignment with labor market demands [14]. The greater the level of information young people have regarding professional opportunities and the labor market's offerings, the more likely they are to make informed career choices [15]. Also, individuals must possess fundamental career-management skills to adeptly assimilate changes into their professional journeys, developing self-efficacy and outcome expectations [3].

It is recommended that each student be offered opportunities for guidance sessions with a career counselor, who may either belong to the school staff or be a specialist external to the institution [15].

Several formative intervention solutions have been developed and validated over time [16]. These interventions have demonstrated that engaging adolescents in activities such as group counseling can enhance skills like career adaptability, resilience, future orientation, perceived efficacy, usefulness, and satisfaction with career construction [16]. These abilities are considered crucial for sustainably navigating future transitions in the job market.

Schools must proactively adapt to the realities, requirements, and needs of the current landscape, addressing the transition from schooling to the labor market or tertiary education in a systematic manner [17,18]. School support is seen as a contextual, timely boundary condition to prepare students to navigate school-to-work and career transitions [3]. Career-ready students are expected to have a well-established career plan, even during high school, with specified goals and short-term, medium-term, and long-term objectives [19,20].

Schools can establish comprehensive career-guidance programs for all students, following the proposal of school management as an institutional policy. Schools should include in their policies a distinct strategy regarding career counseling. This strategy should address providing guidance, preparing graduates for a career, and providing follow-up support [21]. For full guidance services, it is necessary to establish inter-institutional support networks that can provide or assist in specialized activities, giving access to more career contexts and know-how. The cooperation of teachers, parents, and community organizations is vital for the complete preparation of students for their professions. [10,20,22]. Also, it is important for schools to implement tailored counseling policies to prevent discrimination, social inequalities, and disparities [23].

A realistic career plan is founded on self-awareness, which begins with completing the following personal audit: how do you see yourself? [24]. Teachers should guide students in undertaking such an audit [25,26]. Incorporating narrative exercises and goal-setting activities that allow for self-reflection has been identified as a critical component within comprehensive strategies for reaching more positive results [27,28]. Equipping students with the materials they need to make informed decisions about what decent employment entails, encouraging professional curiosity to explore and discover viable options, and shaping adequate representations of the occupational environment are all critical prerequisites for career development [29,30].

Online counseling as a technique for providing career guidance stands out as a viable approach to assisting adolescents in improving their understanding of their employment situations [31]. If this was the only solution accessible during the COVID-19 epidemic, it has the potential to be valued post-pandemic as well, or in a hybrid mix, as the labor market has shifted heavily toward remote working.

In providing career-guidance services to high-school students, schools must examine not only the actual content of such services, the form of delivery, or how to accommodate the institutional support environment, but also the duration of such planned interventions [32]. The details of the concept of such services must be considered contextually, as there are different curricular constructions in different education systems, so the staff delivering them should be specially trained for career counseling, and the career-guidance services must be tailored to the needs of schools and students [33].

### 1.3. Training Teachers for Academic and Career Guidance

Training teachers for career-guidance activities is a solution for empowering students in a rapidly changing society [34]. Similar training initiatives have been found to be beneficial for developing the career-guidance competencies of teachers as career-guidance advisors, provided they receive initial training, information, and preparation for this role [35,36].

Integrating professional development theories, multidisciplinary teamwork, and a post-modern and social-constructivist approach provides a sound theoretical platform for this curricular design [34]. Teachers can play an important role in leading students towards a successful career by developing a closer relationship with their students, providing useful opinions on academic and professional sectors, and helping students feel less pressure while making professional decisions [37]. Teachers also play a crucial role in helping young people shape their career identities and successfully transition into further learning and work [38].

Investing in teacher training for career guidance has significant benefits, including creating an educational environment where students are better prepared to make informed choices and build a successful professional future. Teachers can influence students' career choices by becoming role models while teaching and evaluating their subjects [39,40], making the content more relevant, accommodating long-term goals, and learning from a long-term perspective [39]. Trained teachers can perform career-guidance activities at a basic level in three complementary dimensions of the career-guidance process: personal development, learning, and career planning [8]. Any training for career guidance should include theories related to career development, comprehensive research on occupational aspirations, the evaluation of interests, needs, values, abilities, and other significant factors, occupational categorization, and various sources of occupational information [41].

Investing in the professional development of teachers becomes a strategic approach to foster comprehensive support for individual student success, especially in nations with a student–counselor ratio much larger than the recommended ratio of 1:250 [34]. Furthermore, teachers often do not feel adequately prepared to assume the necessary roles in careers and employability learning without professional development [42].

Additionally, much of the literature on careers and employability learning underscores the significance of integrated whole-school approaches to career guidance [43]. Teacher training should evolve into a holistic school-development initiative rather than a standalone course. Collaboration with universities could explore additional avenues [42], and access to relevant, verified open educational resources.

### 1.4. The Training Concept

Twenty Romanian and twenty Serbian high-school teachers were trained to provide their students with career-guidance activities and a professional path-development program in the autumn of 2021. The teacher training was designed to equip the teachers with the skills needed to conduct comprehensive career-guidance activities across various do-

mains, including academic, professional, and personal development. The training activities aimed to provide teachers with a foundation for an integrated and adaptable approach to career development, ultimately assisting high-school students in developing decision-making and job-search skills for well-informed choices about their academic education and career paths. The training content focused on topics such as "self-knowledge and personal development", "the job market and its dynamics", and "career planning" [44]. The selected topics were aligned with the needs of high-school students for academic, personal, and career growth and development [19,24,30].

The training was implemented through a learning-management platform (LMS), with all the teaching activities being transferred online due to COVID-19 restrictions. The LMS served as a platform for communication, interaction, and the dissemination of scientific knowledge in the field. It facilitated a connection among experts, trainers, and career counselors to guide teachers throughout the career-guidance activities.

In addition to the 50 h of online training on the fundamentals of CAC, all activities were organized and made openly accessible through a comprehensive career-guidance handbook [44]. A joint team with both Romanian and Serbian specialists worked together to design the career-guidance and supporting materials that teachers used to conduct counseling activities. School counselors were mentors for the teachers; they offered permanent in-person support, while university experts provided on-demand assistance through the platform. At the end of each month, an online debriefing activity was conducted, allowing teachers to share their experiences, receive feedback, and plan for the upcoming month's events.

## 2. Methodology

### 2.1. Research Question

What are the teachers' perceptions of the effectiveness of the career-guidance training program?

### 2.2. Research Design

To address the research question, a mixed methodology was employed. Initially, 20 high-school teachers from a Romanian high school and 20 high-school teachers from two Serbian high schools participated in the career-guidance training program. Over a two-month period, from August to September 2021, these teachers engaged in vocational and career-development activities, applying the acquired knowledge under the supervision of course trainers. The two-month teacher training in career guidance concluded with participants responding to a questionnaire designed to assess their opinions, perceived difficulties, and learning outcomes.

This research assessed the cross-national training program's effectiveness in enhancing teachers' competence in career guidance. The effectiveness assessment employed a mixed-research design, combining quantitative data from a questionnaire with qualitative insights obtained through open-ended questions. The assessment of the training's effectiveness was conducted by examining the participants' opinions of the course following its completion. The effectiveness and long-term viability of the training were assessed using a subsequent survey that consisted of open-ended questions and was administered 24 months later. The training's effectiveness and sustainability were measured both at the level of the entire group of participants and, comparatively, between the Romanian and Serbian participants. Are there differences in teachers' perceptions about the training's effectiveness depending on their nationality?

The following hypotheses were advanced:

**H1.** *The majority of participant teachers appreciated the course content and rated the delivery as very good.*

We anticipated a high level of teacher satisfaction due to the course's design by a cross-national team of experts in the field of career counseling, and the way it was run, encompassing various types of support. The course and project idea also garnered approval and financial support from the project-evaluation committee.

**H2.** *The majority of participant teachers perceived the design and implementation of career-guidance activities as challenging.*

We hypothesized that teachers would perceive the design and implementation of counseling activities as challenging, given their professional training in specializations other than education, psychology, or counseling. The participants taught subjects such as mathematics, foreign languages, and geography, etc. Additionally, none of the teachers had undergone prior counseling training.

**H3.** *The majority of participant teachers perceived average learning outcomes.*

This assumption is grounded in the expectation that teachers may possess a low self-efficacy belief in their guidance competence. Therefore, it is assumed that they anticipate learning outcomes at, at most, an average level.

### 2.3. Participants

Forty high school teachers, with twenty from Romania and twenty from Serbia, took part in both the career-guidance training and this research. In the initial phase, 20 Serbian and 20 Romanian teachers completed the impact questionnaire. From this group, 11 Romanian teachers and 12 Serbian teachers provided responses to the sustainability questionnaire 24 months later.

The participating teachers from both countries are subject-based specialist teachers in mathematics, foreign languages, economics, geography, physics, or history. The participants possess teaching experience ranging from 5 to 30 years, with an average of 22.52 years and a standard deviation of 8.33 years. None of them have previously completed a course in career guidance.

### 2.4. Research Instruments

Two questionnaires were administered, one to measure the effectiveness of the training and another to evaluate the sustainability of the training.

The questionnaires were translated bilingually into both Romanian and Serbian. All responses were provided in both languages and subsequently translated into Romanian for statistical and content analysis.

The questionnaire was distributed online at the end of the training and comprised 15 multiple-choice questions and two open questions [45]. The responses to the 15 multiple-choice questions were recorded on a 5-point Likert scale, where a rating of 5 corresponds to 'very good' and a rating of 1 corresponds to 'very bad'. The teachers completed the short-term evaluation questionnaire in the Google Forms design. The questions were developed to identify the teachers' opinions on the course content and delivery methods, on the difficulties that they faced in designing and implementing the counseling activities, and on their perceptions of the learning outcomes. The approval of the research ethics body of our university was received for the research undertaken.

The "sustainability" questionnaire, run 24 months after completing the training, comprised six questions, including two multiple-choice items and four open-ended queries. Its purpose was to collect comprehensive data concerning the counseling activities undertaken by participant teachers over the long term, 2-year period following the completion of the course. The questions specifically addressed the grades in which teachers implemented career-guidance activities, the covered topics, the applied knowledge, the counseling skills

utilized, and the experiences encountered. The teachers' answers underwent content analysis, with several themes emerging for each question, as presented in the results section.

## 3. Results

### *3.1. The Training Program's Effectiveness*

To evaluate the effectiveness of the career-guidance training, the responses from the first questionnaire, completed at the conclusion of the course, were subjected to statistical analysis using the JASP statistical software program [46]. The participating teachers utilized a 5-point Likert scale to express their level of satisfaction regarding the course content, trainers, the course's utility, the accessibility of the learning platform, the practical aspects of the applications, and the didactic materials. The responses were analyzed in terms of frequencies and percentages and are presented in Table 1.

**Table 1.** Teachers' level of satisfaction about the training program.

|  | Content | | Trainers | | Course's Utility | | LMP Accessibility | | Applications | | Teaching Materials | |
|---|---|---|---|---|---|---|---|---|---|---|---|---|
|  | **RO** | **SRB** | **RO** | **SRB** | **RO** | **SRB** | **RO** | **SRB** | **RO** | **SRB** | **RO** | **SRB** |
| 5 | 45% | 75% | 65% | 95% | 60% | 75% | 35% | 40% | 45% | 55% | 40% | 80% |
| 4 | 50% | 20% | 25% | 5% | 35% | 25% | 45% | 40% | 45% | 45% | 50% | 20% |
| 3 | - | 5% | 5% | - | - | - | 10% | 5% | 5% | - | 5% | - |
| 2 | 5% | - | 5% | - | 5% | - | 10% | 5% | 5% | - | 5% | - |
| 1 | - | - | - | - | - | - | - | 10% | | - | - | - |

Note: 5—Excellent, 4—Very good, 3—Good, 2—Satisfactorily, 1—Unsatisfactory, RO—Romanian teachers; SRB—Serbian teachers.

Upon analyzing the data obtained through frequency analysis, it is evident that the hypothesis has been affirmed. A majority of participating teachers consistently rated the course as either excellent or very good across all dimensions, including content, trainers, didactic materials, applications, learning-management systems (LMS), and overall utility. The trainers received the highest level of appreciation, while the LMS was deemed the least appreciated dimension.

Although there are no significant differences in the perceptions of Romanian and Serbian teachers, a slight inclination towards a higher appreciation is noticeable among Serbian teachers (Table 2). All chi-square tests conducted yielded non-significant results (*p*-value varying from 0.06 to 0.53), indicating no significant differences between the perceptions of the Romanian and Serbian teachers for the variables analyzed.

**Table 2.** Chi-square analysis of the teachers' level of satisfaction about the training program. Romanian vs. Serbian Teachers.

|  | Content | | Trainers | | Course's Utility | | LMP Accessibility | | Applications | | Teaching Materials | |
|---|---|---|---|---|---|---|---|---|---|---|---|---|
|  | $\chi^2$ | *p* | $\chi^2$ | *p* | $\chi^2$ | *p* | $\chi^2$ | *p* | $\chi^2$ | *p* | $\chi^2$ | *p* |
| RO-SR | 2.17 | 0.33 | 5.79 | 0.12 | 1.66 | 0.43 | 2.79 | 0.52 | 2.2 | 0.53 | 7.23 | 0.06 |

Note: $\chi^2$—chi square test, *p*—probability value (*p*-value), RO—Romanian teachers; SR—Serbian teachers.

In conclusion, it can be asserted that this training was meticulously designed and delivered in a scholarly manner. It successfully proposed and conveyed pertinent knowledge in the field of career guidance, earning commendations from the participating teachers.

Table 3 illustrates the teachers' perceptions regarding the utility of various career-guidance topics presented. All topics are deemed highly useful and pertinent, with scores exceeding 75%.

**Table 3.** Perceived usefulness of career-counseling topics addressed in the training.

| | The Labor Market | | Personal Development | | Communication and Social Relations | | Learning Management | | Career Planning | | Lifestyle | |
|---|---|---|---|---|---|---|---|---|---|---|---|---|
| | **RO** | **SRB** | **RO** | **SRB** | **RO** | **SRB** | **RO** | **SRB** | **RO** | **SRB** | **RO** | **SRB** |
| 5 | 85% | 80% | 85% | 90% | 85% | 80% | 80% | 75% | 75% | 90% | 85% | 55% |
| 4 | 15% | 20% | 15% | 10% | 15% | 20% | 20% | 25% | 25% | 10% | 15% | 45% |
| 3 | - | - | - | - | - | - | - | - | - | - | - | - |
| 2 | - | - | - | - | - | - | - | - | - | - | - | - |
| 1 | - | - | - | - | - | - | - | - | - | - | - | - |

Note: 5—Very important, 4—Important, 3—Neutral, 2—Unimportant, 1—Very unimportant, RO: Romanian teachers; SRB: Serbian teachers.

An open-ended question also sought feedback from teachers regarding the most interesting and well-received topics within the implemented career-guidance activities in both individual and group counseling. The most frequently mentioned topics were consistent among both Romanian and Serbian teachers, focusing on creating a CV, crafting a letter of intent, presenting oneself effectively in an interview, and developing a career plan. This observation indicates that, for high-school students, the primary emphasis lies not only in self-awareness and making appropriate career decisions but particularly in the practical aspect of searching for and securing employment. Consequently, it is recommended that activities addressing these aspects be incorporated more prominently into the career-guidance curriculum.

A chi-square test was conducted to determine if there are differences in the perceptions of Romanian and Serbian teachers in the perceived usefulness of career-counseling topics (Table 4). All chi-square tests conducted yielded non-significant results ($p$-value varying from 0.08 to 0.70), indicating no significant differences between the perceptions of the Romanian and Serbian teachers for the variables analyzed.

**Table 4.** Chi-Square analysis of the teachers' perceived usefulness of career-counseling topics. Romanian vs. Serbian Teachers.

| | The Labor Market | | Personal Development | | Communication and Social Relations | | Learning Management | | Career Planning | | Lifestyle | |
|---|---|---|---|---|---|---|---|---|---|---|---|---|
| | $\chi^2$ | $p$ | $\chi^2$ | $p$ | $\chi^2$ | $p$ | $\chi^2$ | $p$ | $\chi^2$ | $p$ | $\chi^2$ | $p$ |
| RO-SR | 0.53 | 0.46 | 0.22 | 0.63 | 0.17 | 0.67 | 0.14 | 0.70 | 1.55 | 0.21 | 3.28 | 0.08 |

Note: $\chi^2$—chi square test, $p$—probability value ($p$-value), RO—Romanian teachers; SR—Serbian teachers.

*3.2. Difficulties in Designing and Implementing Career-Guidance Activities*

The perceived difficulty in designing and implementing individual and group guidance activities was also calculated in percentages for both Romanian and Serbian groups (see Table 5).

**Table 5.** Perceived difficulties in designing and implementing career-guidance activities.

| | Designing Individual Guidance Activities | | Designing Group Guidance Activities | | Implementing Individual Guidance Activities | | Implementing Group Guidance Activities | |
|---|---|---|---|---|---|---|---|---|
| | **RO** | **SRB** | **RO** | **SRB** | **RO** | **SRB** | **RO** | **SRB** |
| 5 | - | - | - | - | - | - | - | - |
| 4 | 50% | 77% | 44% | 83% | 55% | 60% | 50% | 60% |
| 3 | 22% | 23% | 23% | 17% | 17% | 40% | 22% | 40% |
| 2 | 28% | - | 28% | - | 16% | - | 22% | - |
| 1 | - | - | 5% | - | 12% | - | 6% | - |

Note: 5—Very easy, 4—Easy, 3—Neutral, 2—Difficult, 1—Very difficult; RO: Romanian teachers; SRB: Serbian teachers.

The results indicated that Serbian teachers found it easier than Romanian teachers to design and implement both group and individual guidance activities. When comparing individual to group activities, individual activities were perceived as slightly easier to design and implement. In the comparison between designing and applying, the design was found to be slightly more challenging than the implementation. Overall, the majority of teachers in both groups found it easy to both design and implement guidance activities, infirming confirming Hypothesis 2.

A chi-square test was conducted to determine if there are differences in the perceptions of Romanian and Serbian teachers regarding perceived difficulties in designing and implementing career-guidance activities (Table 6). All chi-square tests conducted yielded significant results (*p*-value varying from 0.00 to 0.002), indicating that the Serbian teachers found it significantly easier than Romanian teachers to design and implement both group and individual guidance activities.

**Table 6.** Chi-square analysis of the perceived difficulties in designing and implementing career guidance activities: Romanian vs. Serbian Teachers.

|  | Designing Individual Guidance Activities | | Designing Group Guidance Activities | | Implementing Individual Guidance Activities | | Implementing Group Guidance Activities | |
|---|---|---|---|---|---|---|---|---|
| RO-SR | $\chi^2$ | $p$ | $\chi^2$ | $P$ | $\chi^2$ | $p$ | $\chi^2$ | $p$ |
|  | 12.47 | 0.002 | 14.64 | 0.002 | 14.40 | 0.002 | 15.85 | 0.00 |

Note: $\chi^2$—chi square test, *p*—probability value (*p*-value), RO—Romanian teachers; SR—Serbian teachers.

### 3.3. Perceived Learning Outcomes

Students' learning outcomes were operationalized across three constructs: the level of involvement of high-school students in activities, students' didactic and interpersonal communication, and students' success in the qualitative achievement of didactic tasks (see Table 7).

**Table 7.** Perceived learning outcomes in individual and group career-guidance activities.

|  | Students' Involvement in GGA | | Students' Communication in GCA | | Students' Success in GCA | | Students' Involvement in IGA | | Students' Communication in IGA | | Students' Success in IGA | |
|---|---|---|---|---|---|---|---|---|---|---|---|---|
|  | RO | SRB | RO | SRB | RO | SRB | RO | SRB | RO | SRB | RO | SRB |
| 5 | 40% | 44% | 55% | 55% | 39% | 39% | 44% | 44% | 44% | 44% | 27% | 21% |
| 4 | 55% | 50% | 45% | 45% | 61% | 61% | 66% | 66% | 66% | 66% | 67% | 73% |
| 3 | 5% | 6% | - | - | - | - | - | - | - | - | 6% | 6% |
| 2 |  | - | - | - | - | - | - | - | - | - | - | - |
| 1 | - | - | - | - | - | - | - | - | - | - | - | - |

Note: 5—Excellent, 4—Very good, 3—Good, 2—Satisfactorily, 1—Unsatisfactory, RO: Romanian teachers; SRB: Serbian teachers; GGA: group guidance activities; IGA: individual guidance activities.

The results of the percentage analysis indicate that teachers perceived these learning outcomes positively and very positively, both in the guidance activities conducted individually and in groups. The results are almost identical for both Romanian and Serbian teachers. Hypothesis 3 is not confirmed; the satisfaction level with learning outcomes is above average.

### 3.4. Sustainability of the Career-Guidance Training

To assess the sustainability of acquired competencies and the continued application of learned skills by teachers, a follow-up questionnaire was administered 24 months after the completion of the training. Out of the 20 Romanian and 20 Serbian teachers, 11 Romanians and 12 Serbians responded to this questionnaire.

In addition to the three open-ended questions, which will be analyzed further below, inquiries were made regarding the class where guidance activities were implemented, the content of these activities, and the role undertaken by the teachers, either as class teachers or classroom instructors.

All the responding teachers engaged in career-guidance activities, predominantly with students in the final grades, specifically the 11th (36.4% RO; 25% SRB) and the 12th (45.5% RO; 50% SRB). The majority of teachers conducted career-guidance activities in the capacity of class teachers (63.6% RO; 16.7% SRB), while others served as teachers instructing the respective classes (18.2% RO; 75% SRB).

Concerning the topics most frequently addressed in career guidance during this period, a shift can be observed compared to those with the most success during the implementation period, as presented above. While topics related to job searching (CV, letter of intent, interview presentation) attracted teachers and students during the training period, teachers now declare that they most frequently engaged in counseling activities regarding the choice of profession, self-knowledge for informed decision making, or the presentation of different types of professions.

A plausible explanation for this phenomenon is that a majority of the guidance activities were conducted in the twelfth grade, where students are in the immediate situation of making a career choice, making topics related to career decisions more relevant and useful. However, students in the final year have likely already made their career choices, and their needs may differ, necessitating different types of career-guidance activities.

A deductive content analysis was conducted using three open-ended questions, each corresponding to one of the three predetermined themes of analysis: guidance knowledge, skills, and attitudes.

### 3.4.1. Knowledge

When asked about the crucial knowledge that a counselor must possess, all the teachers' provided examples of topics within the realm of career counseling, including career awareness, self-knowledge, and career planning. 'There are several topics that I covered with the students, and they helped me in class. For example: how to identify students' strengths, how to set development goals or how to make a career plan' (P 7, RO).

Only one–two teachers mentioned knowledge about guidance activities, such as the traits of adolescent development or guidance methods and techniques. 'What I learned from this course, which was practically verified in class, is that the children are interested and willing to cooperate with the teacher, learn something new, and acquire new skills that they will definitely need later in life. The knowledge about adolescent psychology, communication in difficult situations, or conflict resolution helped me in particular' (P. 3, SRB).

### 3.4.2. Skills

Among the guidance skills, those most frequently mentioned included empathy, closeness to the students, effective communication, and teamwork. 'Communication skills and empathy. Getting close to students and getting to know them first hand and their family circumstances and life experience contributes in many ways to realistic assessment of students and their abilities' (P. 10, SRB).

Some teachers doubt whether a teacher, even when trained, possesses the same level of skills necessary for career guidance as a specialized counselor. 'I don't think that a full-time teacher, possibly paid by the hour, can compensate for the activity of a counselor, although as a class teacher he is put in a position to advise students on a lot of problems. This course was beneficial because it provided materials and information that can guide you in the classroom. As for skills, I think that the teacher's communication ability is essential' (P. 6, RO).

### 3.4.3. Attitudes

The most frequently mentioned attitudes included an increase in confidence to conduct career-guidance activities, a heightened sense of responsibility towards students, and an enhanced willingness to support students in various aspects, extending beyond issues related to career guidance. 'After finishing the course, I carried out individual- and group-counseling activities. The open attitude of the students, the desire to discover themselves and make the right career choice, made me realize the great need for support they have not only in choosing a career but also in personal problems and I became more available and had a great sense of responsibility'. (P.2, RO).

Some teachers noted a deeper understanding of their students, a closer relationship with them, and the development of qualities beyond those of a counselor, enriching their role as educators. 'All this experience opened up a new perspective on my role as a teacher: to assist students in their development, not just to instruct them. The accumulated knowledge of guidance helped me in teaching, not only in career counseling' (P.5, SRB).

## 4. Discussions

Career guidance is a crucial service for schools to provide to their students [33]. Making a suitable career choice that offers satisfaction and professional fulfillment is a complex task, often requiring support from adults, teachers, counselors, or parents to make informed decisions [47]. Schools have teachers and specialized counselors in career guidance and should play a central role in this process, ensuring the "constant processes of accompaniment and mediation" ([20], p. 220).

However, there are instances where the demands of career counseling surpass the capacity of existing specialists [4]. The presented training program was designed for teachers who can step in to complement the work of specialized counselors, aimed at enabling the teachers to run career-guidance activities effectively, helping students in their career-development journey. The training concept was rooted in a simplified version of both the Trait-and-Factor Career-Counseling model [48], on the SCCT and on the career-construction theory [3,6,29,39]. Drawing from the Trait-and-Factor model, the training focused on helping teachers understand how individual traits and factors influence career choices, empowering them to assist students in assessing their strengths, interests, and values, while applying basic diagnostic tools. Additionally, by integrating principles from SCCT, the training emphasized the role of self-efficacy beliefs and observational learning in career-related decision making. Based on the observations made on their trainers from university, and on the experiences gathered during the training, teachers learned to foster students' confidence in exploring career options and to provide positive examples of effective career-exploration behaviors. By combining elements of both models, the training aimed to provide a comprehensive framework for teachers to support students in making informed and fulfilling career decisions.

The training involved instructing the teachers to guide high-school students to explore and develop their career interests, identify personal strengths and limitations, enhance self-efficacy, manage expectations, and gain a broader understanding of labor and societal trends, as well as personal and environmental factors. This program aimed to equip 20 Romanian and 20 Serbian teachers with the necessary skills for career guidance, including both an informative component and, significantly, an application and mentoring dimension provided by experts, accompanied by support materials.

The objective of this research was to evaluate the effectiveness of this cross-national training program designed to enhance teachers' competence in career guidance. A longitudinal mixed methodology was conducted. The effectiveness of the program and the sustainability of the acquired skills were assessed 24 months later using two questionnaires containing multiple-choice and open-choice questions. Responses to the questions with multiple choices were analyzed in terms of frequency, while answers to the open-ended questions underwent content analysis.

Firstly, we assessed the effectiveness of the training program. The outcomes revealed that a significant majority of participating teachers consistently evaluated the course as either excellent or very good in all dimensions, including content, trainers, didactic materials, applications, the learning-management system (LMS), and overall course utility. This training was meticulously designed and delivered in a comprehensive manner. It effectively proposed and conveyed relevant knowledge in the field of career guidance, garnering commendations from the participating teachers.

The quality of the training implementation is particularly commendable, considering that it was originally designed for a conventional face-to-face format. However, due to the COVID-19 pandemic, it had to be transitioned to an online format, necessitating swift adaptations of teaching methods [49,50], which proved especially challenging for application-type activities. An essential aspect was the learning-management system (LMS) platform provided to the participants; the success of online learning activities largely hinges on the quality of these platforms. Unfortunately, the LMS recorded the lowest percentage regarding quality, indicating that there is room for improvement in this aspect.

Secondly, the teachers highlighted the challenges they faced in designing and executing counseling activities throughout the training. The findings revealed that Serbian teachers perceived less difficulty than their Romanian counterparts in designing and implementing both group- and individual-guidance activities. When comparing individual and group activities, individual activities were perceived as marginally simpler to design and implement. This result is counterintuitive, as teachers are accustomed to teaching in group situations to classes of students, so group-counseling activities would typically have been perceived as easier to apply. However, participating teachers perceived individual career-guidance activities as slightly simpler to implement, and a possible explanation could be the novelty bias [51], with teachers becoming more involved in the preparation of these activities.

Furthermore, in the comparison between the design and implementation phases, designing was found to be marginally more challenging than the actual implementation. Teachers are unaccustomed to designing counseling activities, which accounts for this result.

Thirdly, the analysis focused on teachers' perceptions of students' learning outcomes. The findings revealed that teachers viewed the learning outcomes in a positive and very positive light, both in individually conducted guidance activities and in group settings. These results exhibited similarity among both Romanian and Serbian teachers.

Twenty-four months after the completion of the training, the sustainability of the acquired competencies was assessed through a content analysis of four open-ended questions. All teachers actively participated in career-guidance activities, primarily with students in their final grades, serving as class leaders or teachers instructing the respective classes.

Over time, there has been a notable shift in the predominant topics addressed in career guidance. Initially, during the training period, topics associated with job search, such as CV preparation, letter-of-intent writing, and interview presentation, captured the attention of teachers and students. However, in the subsequent period, teachers reported a more frequent involvement in counseling activities centered around career choice, self-knowledge for informed decision making, and the presentation of various professions. It is crucial for teachers to analyze students' needs comprehensively and subsequently design and implement activities that align with those needs [4]. Therefore, the direct involvement of students in the planning and design of career-guidance activities at the high-school level is essential [4,7].

When asked about important knowledge for a counselor, all teachers cited topics related to career counseling, such as career awareness, self-knowledge, and career planning. A few teachers did, however, mention knowledge about guidance activities, such as the characteristics of adolescent development or guidance methods and techniques. Consequently, career-guidance training programs should not only focus on content like CVs and letters of intent, which teachers can independently acquire, but also on knowledge directly linked to the guidance process [48]. This encompasses methods and counseling techniques,

assessment tools, decision-making processes, developmental psychology, counseling, and communication skills, along with ethical considerations [48].

In terms of guidance skills, commonly cited attributes encompassed empathy, closeness to students, effective communication, and teamwork. Beyond these recognized skills, training should also cover topics such as active listening, cultural understanding, motivational skills, and flexibility. Moreover, the inclusion of themes related to ethics and confidentiality is crucial and should be integrated into the training program.

The participating teachers frequently reported attitudes including increased confidence in conducting career-guidance activities, heightened responsibility towards students, and a greater willingness to support students beyond career-related concerns. Some teachers also highlighted a deeper understanding of students, closer relationships, and the development of qualities beyond counseling, enriching their role as educators. In conclusion, this training not only enhances guidance skills but also imparts additional qualities and skills applicable to teaching any discipline. Incorporating topics into initial psycho-pedagogical training programs to practice skills like active listening, effective questioning, problem-solving, non-violent communication, and decision making is advisable [35,52].

This career-guidance training has demonstrated its efficiency and sustainability, enhancing the capacity to provide career-guidance services in schools. In the current Romanian context, with a growing demand for career advisors, this training holds potential for replication among other teachers. A research approach should come after the start of new training sessions, evaluating the training requirements for teachers in career guidance, the needs of students making career decisions, the best delivery formats, and the implementation of follow-up procedures.

Despite the many benefits of teachers acting as career counselors, we should mention that it also raises concerns about adding new responsibilities to the teachers' already heavy workload. Adding career-guidance responsibilities without proper compensation could strain their resources and time. While financial compensation would be ideal, it may not always be feasible. Therefore, it is essential for policymakers to implement measures that sustainably support teachers' involvement in career guidance, considering their well-being and professional development.

Possible measures could include providing additional resources and training opportunities specifically tailored to career guidance, offering incentives such as reduced teaching loads or dedicated time for counseling activities, and establishing support networks where teachers can share experiences and seek guidance. Additionally, integrating career guidance into the broader educational curriculum and fostering collaboration between schools and external career-counseling professionals could further alleviate the burden on teachers while enhancing the effectiveness of career-guidance services.

*Limitations*

The primary limitation is the generalizability of the study. This research focused on investigating the effectiveness of a specific training program, and, as such, the results are applicable solely to this program, limiting their generalization. However, there is an intention for program replication among other groups of teachers over time, aiming to accumulate data that may be generalized and to enhance the training based on empirical evidence collected.

The second significant limitation lies in the evaluation of the training effectiveness through a self-administered questionnaire, a subjective method susceptible to the bias of socially accepted answers [45]. To address this limitation for future studies, it is recommended to incorporate more objective measures or triangulating data from multiple sources, such as observation scales completed by an expert or evaluations of students' learning outcomes and satisfaction levels.

The third significant limitation is that the evaluation of the training's impact relied solely on the perspectives of the teachers, neglecting objective data such as the employment rate or enrollment in tertiary education among the young individuals who received guid-

ance. A comprehensive study should incorporate these aspects to provide a more holistic understanding of the training's effectiveness.

A fourth notable limitation of this study is the potential influence of external factors or contextual variables that were not controlled for during the training and evaluation process (such as teachers' specialization, teaching experience, and personality traits), and the meta-analytic study of Sharapova et al. (2023) emphasizes the importance of these variables. These unaccounted-for factors could have impacted the participants' responses and the overall outcomes. Future research should consider a more comprehensive examination of potential confounding variables to enhance the study's validity and reliability.

## 5. Conclusions

This article presents a career-guidance training program designed to equip teachers with the skills to act as career-guidance advisors for high-school students. The study, conducted through two questionnaires comprising multiple-choice and open-ended questions, assesses the training's effectiveness over time. The results indicate high levels of teacher satisfaction and the sustained implementation of learned practices even after two years. Areas for improvement include the learning platform (LMS) and evaluating impact through student feedback. While the study's generalizability is limited, the strength of this paper lies in an exemplified training solution and an illustration of how this can be designed and evaluated, so it can be replicated and adapted to various educational contexts. Prioritizing counseling knowledge and skills, incorporating student input into activity designs, utilizing modern delivery methods, and fostering psycho-pedagogical skills can enhance the effectiveness of similar training programs. Integrating relevant topics into initial psycho-pedagogical training can further enrich teachers' abilities in diverse educational contexts.

Conducting these training programs on a cross-national scale not only addresses specific skills targeted by the training but also facilitates the development of transversal skills. These include cultural understanding, embracing diversity, fostering tolerance, and enhancing collaborative, learning, and teamwork abilities within an international context [49].

In the context of a limited number of career-counseling specialists [4], training teachers to serve as career-guidance advisors emerges as a viable solution. To align the instructional design with societal needs, labor market dynamics, and advancements in the field, it is advisable for training sessions to be coupled with empirical research. This research should focus on assessing the quality, effectiveness, and sustainability of the training and on drawing conclusions grounded in scientific evidence.

**Author Contributions:** Conceptualization, O.S.B., S.S. and A.L., methodology, O.S.B. and A.L.; formal analysis, A.L. and O.S.B.; investigation, O.S.B.; data curation, A.L. and O.S.B.; writing—original draft preparation, O.S.B., O.B. and S.S.; writing—review and editing, O.S.B., A.L., O.B. and S.S.; supervision, S.S. and A.L. All authors made equal contributions to the elaboration of the article. All authors have read and agreed to the published version of the manuscript.

**Funding:** This research was conducted as part of the project IPA RORS 406 for regional trans-border cooperation in Romania–Serbia, "e-Support services for career and vocational counseling of youth entering to the labor market", 2021–2022, coordinated by O.S.B.

**Institutional Review Board Statement:** The study was conducted in accordance with the Declaration of Helsinki, and approved by the Institutional Review Board of West University of Timisoara, protocol no. 86938/20.11.2023.

**Informed Consent Statement:** Informed consent was obtained from all subjects involved in the study.

**Data Availability Statement:** The data that support the findings of this study are available from the corresponding author upon reasonable request.

**Acknowledgments:** The authors would like to thank all participants involved in the "e-Support services for career and vocational counseling of youth entering to the labor market" project.

**Conflicts of Interest:** The authors declare no conflict of interest.

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
