# Peer review of "Training Teachers for the Career Guidance of High School Students"

_education, doi:10.3390/educsci14030289_

Round 1
Reviewer 1 Report (Previous Reviewer 2)
Comments and Suggestions for Authors
p.1.15/16 You write here that the impact and sustainability of the skills is measured albeit only the program is evaluated (there is no effect measure; rating something as excellent does not make it effective). Please revise.
I do find the iniative relevant, but it also raises concerns. Teachers experience a high-workload to begin with and this is adding to that. Reasonable, they need financial compensation if they are properly training. I carefully assume this is (and will not) be the case. This has to be mentioned in the discussion alongside suggestions for the long-term.
p.2.51-57 This is too briefly described. The ties to socio-cognitive theory should be a fouundation for your training, but is not only briefly addressed. I can reason that career guidance is the most crucial albeit using theory to support your main concepts is advised. I am also expecting a reference to this theoretical foundation when you discuss potential solutions, such as training. You do not do this. Why?
p.2.78-83 This is presented in a different font colour.
p.4.185 I do have issues with how your RQ is phrased. The impact and effectiveness implies that you focus on the effect of the training. You briefly mention the learning outcomes, but the majority of the scales focus on evaluation (i.e., satisfaction and perceived usefulness). Up to an extent, one can reason that the latter indicates an impact; however, I would be careful with phrasing it that way. Can you come up with an alternative phrasing?
p.4.185 Impact on what?
p.5.201 I would eventually expect that you also take a look at dropout rates (or engagements scales for students). These consequences are part of the rationale as to why you conduct this research. The measurements do not pertain to those aspects of the "problem". I am expecting that this aspect will be mentioned in the limitations.
p.4.200 You refer here to the effectiveness and long-term viability (see my previous comments about this). Also be consistent in your terminology.
p.5.220 This is an odd hypothesis because this is implicitly addressed in the RQ.
p.5.225 I expect that you also report on the experience from the teacher (you refer to external factors in your limitations).
Looking at the results, you only present descriptives. It is difficult to refer to an impact. There is no t-test or any comparison.
p.8.334 You refer here to sustainability instead impact, effectiveness, and/or viability. Keep your main terminology consistent.
p.8.357 I do not understand the separate subheader.
p.9.386 The order of the references is--according to what I know from this reference format--incorrect. Please check.
Limitations
p.11.481 I would suggest to use the header "Limitations" (Limits is unusual).
The suggestion for triangulation is a step in the right direction, but I still miss the link to the main consequences of the lack of guidance: drop out, lack of engagement, etc. You focus strongly but solely on teachers.
p.11.497 You refer to "their role" but it is unclear to what you are referring.
References
Still contain inconsistencies. Please check again (e.g., compare 6 with 8).
Comments on the Quality of English LanguageSee above.
Author Response
Dear Reviewer,
Thank you so much for your helpful feedback. Please find attached our answers.

Reviewer 2 Report (New Reviewer)
Comments and Suggestions for Authors
The paper says that it is informed by social cognitive career theory, but this perspective is not strongly apparent in the rest of the paper. The focus on the use of narrative career counselling suggests another theoretical perspective (e.g. career construction theory). This theoretical uncertainty needs to be sorted out prior to publication.
The section summarising career guidance for high school students reads as a bit of a random list. I would suggest looking at some papers that provide an overview of career guidance in schools in different contexts. E.g.
Gatsby Charitable Foundation. (2014). Good career guidance. Gatsby Charitable Foundation.
Hughes, D. (2017). Careers work in England’s schools: politics, practices and prospects. British Journal of Guidance & Counselling, 45(4), 427-440.
Sharapova, N., Zholdasbekova, S., Arzymbetova, S., Zaimoglu, O., & Bozshatayeva, G. (2023). Efficacy of School-Based Career Guidance Interventions: A Review of Recent Research. Journal of Education and e-Learning Research, 10(2), 215-222.
There is some important work which explicitly addressed the role of teachers in relation to career guidance that has been missed out. See
Andrews, D., & Hooley, T. (2017). ‘… and now it’s over to you’: Recognising and supporting the role of careers leaders in schools in England. British Journal of Guidance & Counselling, 45(2), 153-164.
Andrews, D., & Hooley, T. (2019). Careers leadership in practice: a study of 27 careers leaders in English secondary schools. British Journal of Guidance & Counselling, 47(5), 556-568.
Dodd, V., & Hooley, T. (2018). The development of the teachers’ attitudes toward career learning index (TACLI). Teacher Development, 22(1), 139-150.
Hooley, T., Dodd, V., & Shepherd, C. (2016). Developing a new generation of careers leaders: An evaluation of the Teach First Careers and Employability Initiative. Teach First
Hooley, T., Watts, A. G., & Andrews, D. (2015). Teachers and careers: The role of school teachers in delivering career and employability learning. Teach First.
Karacan Ozdemir, N., Akçabozan Kayabol, N. B., Aydın, G., & Tatlı, C. E. (2022). Fostering teachers’ career education competencies: test of a training programme. British Journal of Guidance & Counselling, 50(3), 462-473.
Kuijpers, M., & Meijers, F. (2017). Professionalising teachers in career dialogue: An effect study. British Journal of Guidance & Counselling, 45(1), 83-96.
Wong, L. P., Yuen, M., & Chen, G. (2022). Career guidance and counselling: the nature and types of career-related teacher social support in Hong Kong secondary schools. British Journal of Guidance & Counselling, 50(6), 897-915.
I do wonder how much focus should be given to the quantitative results given the population and response rates were so low. I would be inclined to slim down this section of the paper and give more space to the qualitative data that was collected. At present the qualitative data is poorly presented with no clear approach to coding offered. It would be good to see how the data were coded and to see more quotes to allow the voices of the participants to come through more.
In the discussion it would be good to hear more about the challenges that teachers face. At the moment this is really just an evaluation of an intervention. My question is, is this kind of training sufficient or is there a need to address the conditions of work of the teachers and to transform the schools in which they work in. Also what is the role that the school counsellor plays in this activity?
Comments on the Quality of English LanguageThe English is mainly of a high standard. Some proofing is needed.
Author Response
Dear Reviewer, Thank you so much for your feedback. Please find attached our answer.

Round 2
Reviewer 2 Report (New Reviewer)
Comments and Suggestions for Authors
I agree that the authors have completed all suggested amendments.
I think that the paper is now ready to publish. I would suggest a final proofing review, but other than that I think that it is good to go.
Comments on the Quality of English LanguageIt is fine. Minor proofing needed
This manuscript is a resubmission of an earlier submission. The following is a list of the peer review reports and author responses from that submission.
Round 1
Reviewer 1 Report
Comments and Suggestions for Authors
This manuscript highlights the importance of leveraging teachers to serve as career counselors and presents a model that can be helpful in school settings.
The Introduction is clear and sets up the rationale for the study, but needs a more clearly defined hypothesis.
The method is fairly straight forward but there needs to be more detail with respect to the qualitative analysis method that was employed as there are a number of methods to engage in qualitative data analysis.
The results section needs substantantial work in detailing how the analysis was carried out and presenting the data in a consistent manner. The charts in the manuscript do not have uniform font sizes. I would recommend using similar research articles to model the formatting and presenting qualitative findings.
Comments on the Quality of English Language
1. There are a few places where the manuscript is written in future tense, probably from the original proposal. These need to be changed to the past tense.
2. The authors should conduct a thorough review of the language usage as there are some sentences that need revision.
Reviewer 2 Report
Comments and Suggestions for Authors
Abstract
"to work or academia" = and/or (to be more nuanced and inclusive).
Avoid using "thus" in the middle of the sentence. Moreover, the informal density is pretty high in that sentence (you want to convey too much in one sentence). I would suggest to split this sentence in two sentences.
Do you refer to researchers and teachers as part of academia and policymakers as part of "work"? I do not understand why you only list these three.
You refer to "high school students" and then to "kids"? Use terminology suitable for the educational context. This means you refer to students (or more specifically to high school students).
"with the challenges of the shirt" = of that shift? Or of this shift? You need to be more specific.
"based on the abilities learned" = Such as? Up till now, I still do not know to what skills and/or knowledge you refer.
If you are using APA 7, please place the N is italics.
"feedback provided by the teachers at the conclusion" = I do not know what you mean with "at the conclusion". This is uncommon (and not appropriate language use in this context).
"in running the counselling activities" = you refer to running as managing, monitoring, executing? Be specific. The rest of the sentence makes it grammatically incorrect (from "the counselling...").
"to cope even better with the task of academic..." = The task of academic? What does this mean?
The terminology in the last sentence of your abstract is different than the terminology used in the sentences before that.
Introduction
The first sentence in this section requires references (plural). You highlight how much has been done. Your references should resemble this.
Career and academic counselling are apparently two different concepts (you make this explicit in the second sentence of the introduction).
"instructors who feel unqualified for it" = Because they are. They are taught to instruct (not to councel and/or guide students). Furthermore, are instructors the same as class teachers? Again, your terminology is inconsistent. As a result, it is confusing for the reader.
"at the conclusion of" = Odd. Please revise.
The statement "Teachers frequently avoid..." requires a source.
"dedicated continuing professional development" = Comes as a surprise. You also refer to academic and career guidance demands. This is illogical for readers. This paragraph is lacking coherence.
"dedicated training they received" = Who are "they"? You refer to teachers (?) in the previous paragraph, but you need to repeat that here.
"high-school" is written differently compared to the abstract. In addition, the context of high school hasn't been properly introduced.
"they ran counselling activities" = Who are they? I also do not know if this is part of the methodology or not?
"a better chance to make informed decisions about their..." = Thus, the goal is to support critical thinking and decision-making?
You refer to a new (?) concept here as well: "academic and professional prospects".
To what "special incentives" are you referring?
"The intervention" = What intervention?
Is "deficit" the correct term? Is "lack of" more appropriate?
The aim of having at least one specialist counsellor" comes as a surprise (this hasn't been introduced properly).
The insertion of reference 2 (on page 2) is odd with the parenthesis.
The statement "Youth unemployment in academic..." requires a source.
"According to studies" = This has to be addressed more specifically. You make a very bold generalization here that is not justified.
Why do you highlight the role of class principals? This comes as a surprise as well.
The current study does not reflect upon something; you do this as authors.
Exposure to activities is too vague. Because I also assume you want students to engage with them to solidify the skills and/or knowledge.
"fairly limited" = Vague.
A hyphen is not the same an as em dash.
The transition to "Everything, according to..." is unclear.
What do you mean with "formative intervention solutions"? Would intervention be suffice in this sentence? Your description and/or word choice tend to overcomplicate things.
The text after reference 6 has been placed in a different colour.
The transition to the paragraph about education and training systems does not follow logically. What are training systems? I have never heard of this term.
On page 3, you start a paragraph with "school" whereas you discussed education before. Keep essential terms consistent.
"with the help of a creative principal" = Earlier you used a different "principal".
With the amount of feedback I had to provide on these two (three) pages, I would suggest to go over your work and apply the feedback to the remainder of the manuscript as well. This version is unsuitable for publication.
Comments on the Quality of English LanguageSee above.